# Anti-Leukemic Properties of Aplysinopsin Derivative EE-84 Alone and Combined to BH3 Mimetic A-1210477

**DOI:** 10.3390/md19060285

**Published:** 2021-05-21

**Authors:** Sungmi Song, Sua Kim, Eslam R. El-Sawy, Claudia Cerella, Barbora Orlikova-Boyer, Gilbert Kirsch, Christo Christov, Mario Dicato, Marc Diederich

**Affiliations:** 1Department of Pharmacy, College of Pharmacy, Seoul National University, 1 Gwanak-ro, Gwanak-gu, Seoul 08626, Korea; sson35@snu.ac.kr (S.S.); suakim@snu.ac.kr (S.K.); claudia.cerella@lbmcc.lu (C.C.); barbora.orlikova@lbmcc.lu (B.O.-B.); 2Chemistry Department of Natural Compounds, National Research Centre, Dokki, 12622 Giza, Egypt; eslamelsawy@gmail.com; 3UMR CNRS 7565 SRSMC, Université du Lorraine, 57070 Metz, France; gilbert.kirsch@univ-lorraine.fr; 4Laboratoire de Biologie Moléculaire et Cellulaire du Cancer, Hôpital Kirchberg, 9, Rue Edward Steichen, 2540 Luxembourg, Luxembourg; dicato.mario@chl.lu; 5Service d’Histologie, Faculté de Médicine, Université de Lorraine, INSERM U1256 NGERE, 54000 Nancy, France; christo.christov@univ-lorraine.fr

**Keywords:** aplysinopsin analogs, indole alkaloids, marine source, chronic myeloid leukemia, BH3 mimetics

## Abstract

Aplysinopsins are a class of marine indole alkaloids that exhibit a wide range of biological activities. Although both the indole and N-benzyl moieties of aplysinopsins are known to possess antiproliferative activity against cancer cells, their mechanism of action remains unclear. Through in vitro and in vivo proliferation and viability screening of newly synthesized aplysinopsin analogs on myelogenous leukemia cell lines and zebrafish toxicity tests, as well as analysis of differential toxicity in noncancerous RPMI 1788 cells and PBMCs, we identified EE-84 as a promising novel drug candidate against chronic myeloid leukemia. This indole derivative demonstrated drug-likeness in agreement with Lipinski’s rule of five. Furthermore, EE-84 induced a senescent-like phenotype in K562 cells in line with its cytostatic effect. EE-84-treated K562 cells underwent morphological changes in line with mitochondrial dysfunction concomitant with autophagy and ER stress induction. Finally, we demonstrated the synergistic cytotoxic effect of EE-84 with a BH3 mimetic, the Mcl-1 inhibitor A-1210477, against imatinib-sensitive and resistant K562 cells, highlighting the inhibition of antiapoptotic Bcl-2 proteins as a promising novel senolytic approach against chronic myeloid leukemia.

## 1. Introduction

Chronic myeloid leukemia (CML) is characterized by the oncogenic *BCR-ABL1* fusion gene expression, which codes for a leukemogenic tyrosine kinase [1]. The first-line therapy for CML is imatinib, a tyrosine kinase inhibitor (TKI) that selectively inhibits the activity of the BCR-ABL fusion protein. Since the discovery of imatinib, the overall survival rate of patients with CML has drastically increased, with patients showing durable responses after imatinib treatment [2]. Leukemia cells, however, develop resistance mechanisms to escape chemotherapy; therefore, despite the high remission rate, a significant number of patients develop resistance or become intolerant to imatinib treatment. Furthermore, 33% of patients who receive imatinib treatment do not achieve a complete cytogenetic response (CCyR) [3]. Thus, alternative strategies for the management of CML are needed to combat chemoresistance, for which compounds of natural origin have shown promise as potential therapeutic agents by their capacity to induce cellular stress mechanisms sensitizing leukemia cells against cytotoxic treatments.

The endoplasmic reticulum (ER) is responsible for protein translocation, proper folding, and protein post-translational modifications [4]. Altered cell metabolism and inflammation may disrupt this balance and result in ER stress that can trigger the unfolded protein response (UPR) [5]. The UPR represents a series of adaptive cellular mechanisms designed to restore protein homeostasis [2], and ER stress activates apoptotic cell death under severe or chronic stress conditions [6]. Autophagy is also a stress-induced cell survival program that involves a catabolic process to degrade large protein aggregates and damaged organelles in autophagosomes [7]. Although autophagy and ER stress function independently, increasing evidence supports that these processes can be coactivated [8].

Aplysinopsin and its derivatives possess rich structural diversity and have been reported to exhibit a wide range of medicinal and biological activities. For example, they act as neuromodulators [9] and possess antineoplastic [9], antiplasmodial [10], and antimicrobial activities [11]. Interestingly, aplysinopsins display cytotoxicity against a range of cancer cell lines [12]. However, their anticancer potential in leukemic cell lines and the corresponding molecular mechanisms remain to be further investigated.

Here we evaluated the anti-leukemic activity of aplysinopsin (EE-115) and analogs EE-31, EE-80, EE-84, and EE-92 (Scheme 1) against chronic myeloid leukemia cell lines. EE-84 exhibited drug-like properties in line with Lipinski’s rule of five and showed a stronger cytostatic and cytotoxic effect on leukemia cells than healthy cell models. Furthermore, its safety profile was validated in vivo by using developing zebrafish larvae. Mechanistically, EE-84 induced a senescent-like phenotype in line with its cytostatic activity, triggered autophagy, ER stress, metabolic alterations, and mitochondrial dysfunction. In addition, EE-84 sensitized imatinib-sensitive and -resistant K562 cells against the Mcl-1 inhibitor A-12101477 to induce caspase-dependent apoptosis. Altogether, this study warrants further investigation of the aplysinopsin analog EE-84 as a preclinical drug candidate against chronic myeloid leukemia.

## 2. Results

### 2.1. Aplysinopsin Analogs Display Cytostatic Activities in Myeloid Leukemia Cells

Aplysinopsin (EE-115) and its analogs EE-31, EE-80, EE-84, and EE-92 (Figure 1) were tested for their anti-leukemic effects on the myeloid leukemia cell line K562, using the trypan blue exclusion test (Table 1 and Table 2 and Appendix A). NMR spectrum data of ^1^H of the compounds EE-31, EE-80, EE-84, EE-92, and EE-115 (Appendix A) and ^13^C NMR spectrum data of the compounds EE-31, EE-80, EE-84, and EE-92 (Appendix A) were provided.

Compounds EE-31, EE-80, EE-92, and EE-115 only weakly affected the overall growth (Appendix A) and viability (Appendix A) of K562 cells at concentrations up to 50 µM. Interestingly, EE-84 inhibited the growth of K562 with a GI_50_ of 32.22 ± 3.91 μM and 19.07 ± 0.80 μM after 48 and 72 h, respectively (Appendix A).

Based on the GI_50_ results in the K562 cell line, we selected EE-84 for further investigations compared to EE-115, the parental compound. To generalize the antiproliferative effect of EE-84 in CML, GI_50_ values of EE-84 were calculated on imatinib-sensitive (KBM5, MEG01) and -resistant (IR) (K562IR and KBM5IR) CML cells (Table 1). We next validated the cytostatic profiles of EE-84 by Methocult colony formation assays (CFA) to assess the compounds’ effects on the clonogenic potential in a 3D culture environment (Figure 2A–E). EE-84 reduced the total surface area and average size of K562 colonies after a 48-h pretreatment (Figure 2A). Similar results were observed for other CML cell lines (Figure 2A–E).

Considering the moderate cytotoxic effect of 50 μM EE-84 and EE-115 at 72 h in K562 cells (Appendix A), we quantified cell death induction using Hoechst/PI staining by fluorescent microscopy (Figure 3A). EE-84 reduced K562 cell viability leading to 11.0 ± 4.23% of propidium iodide (PI)-positive cells without nuclear fragmentation. A similar percentage of cell death was shown after quantification of Annexin V APC/PI by FACS (Figure 3B). EE-84 induced approximately 16.5% cell death (Table 2) at the highest concentration of 50 μM. Imatinib (1 μM) was used as a control. Cell cycle analysis showed an accumulation of cells in the sub-G1 phase of 8.34 ± 1.51% after 48 h and 13.03 ± 1.19% after 72 h with EE-84 (50 µM), concomitant with a reduction of the G1 population from 58.40 ± 2.81% at 48 h to 52.03 ± 2.99% at 72 h (Figure 3C and Table 3). Celecoxib (40 µM) was used as a control.

### 2.2. Drug-Likeness of Compounds EE-84 and EE-115

Next, we examined the drug-likeness of the two compounds according to Lipinski’s rule of five (Table 4) [13]. A molecule is orally active when there are no more than 5 H bond donors and no more than 10 H bond acceptors, when the molecular mass is less than 500 Daltons, and when the octanol–water partition coefficient LogP is not greater than 5. EE-115 did not violate any of the conditions mentioned above. For EE-84, The LogP corresponds to 5.91; however, this aplysinopsin derivative can still be considered a drug-like candidate since one exception is acceptable.

### 2.3. Effect of EE-84 and EE-115 on Healthy Cells and Zebrafish Embryos

We then assessed the effect of EE-84 and EE-115 on healthy models. Even though EE-84 affected the proliferation of RPMI 1788 cells dose-dependently (Appendix A), this modulation was evident at later times (as early as 72 h vs. 48 h in cancer cells), with a higher GI_50_ (40.10 ± 3.79 in RPMI 1788 vs. 19.07 ± 0.80 µM after 72 h in K562 cells; Table 1 and Table 5). Altogether, EE-84 generated significant differential toxicity of RPMI-1788/K562 cells at 40 and 50 µM (Figure 4A). On the other hand, EE-115 inhibited the growth of normal RPMI 1788 cells (Table 5) more than K562 cells (Appendix A). Moreover, EE-84 did not significantly affect the viability of RPMI 1788 cells, whereas EE-115 reduced the viability of RPMI 1788 cells at the highest concentrations (Appendix A). Altogether EE-84 appeared to have a more advantageous safety profile.

We generalized our findings and validated the absence of cytotoxic/cytostatic effects of EE-84 on nonproliferating and proliferating peripheral blood mononuclear cells (PBMCs) from healthy donors. In addition, EE-84 did not induce toxic effects in any of the two cell models (Figure 4B).

We then treated zebrafish embryos 24 h post-fertilization (hpf) to validate the in vivo safety profiles of the aplysinopsin compounds EE-84 and EE-115. The zebrafish were observed after 24 h of exposure to different concentrations of aplysinopsins. No significant morphological changes were notable at the different concentrations of the compounds used (Figure 4C). Concentrations up to 100 µM of EE-84 and EE-115 did not affect the survival rate (Figure 4D) or growth (Figure 4E) of the treated zebrafish. A moderate decrease in the heart rate (Figure 4F) was noted for EE-84.

Altogether, based on these results, we further investigated the molecular and cellular effects of EE-84, considering the observed differential toxicity.

### 2.4. EE-84 Induced Morphological Changes and Cell Stress Responses in CML Cells

EE-84 (50 µM) increased the cell size (Figure 5A) and the number of intracellular vacuoles up to 96 h. These morphological changes were quantified by flow cytometry (Figure 5B). Compared to DMSO-treated controls, EE-84-treated K562 cells showed a progressive increase in forward scatter values (FSC-H; a.u.) and side scatter (SSC-H; a.u.) as measurements of cell size and granularity, respectively. The increase in cell size and granularity and the cytostatic effects were considered part of a cellular stress response driving cells into senescence. We then examined the senescent-like phenotype using stress-associated (SA)-β-galactosidase staining. We measured increased cell size and SA-β-gal positive cells after treatment with EE-84 at 50 µM for 72 h. Doxorubicin (80 nM), known to induce senescence, was used as a bona fide control (Figure 5C).

In addition, morphological changes were examined by transmission electron microscopy (TEM). K562 cells treated with EE-84 underwent mitochondrial damage after 24, 48, and 72 h of compound exposure as well as mitophagy (Figure 5D). In addition, we assessed the mitochondrial metabolism in K562 after treatment with EE-84 at 30 µM for 72 h by using the Agilent Seahorse XFp Cell Mito Stress test. The oxygen consumption rate (OCR) significantly decreased after treatment with EE-84 compared to control (Figure 5E). Thus, our results confirmed that EE-84 induced cell stress and reduced mitochondrial activity in agreement with morphological alterations of the mitochondria observed by TEM.

### 2.5. Autophagy and Endoplasmic Reticulum (ER) Stress Were Triggered by EE-84 as Cellular Stress Responses in K562 Cells

Autophagy is a catabolic pathway activated in response to different cellular stressors, such as damaged organelles, accumulation of misfolded or unfolded proteins, ER stress, and DNA damage [14]. Based on the TEM results, we then investigated the expression levels of the microtubule-associated protein-1 light chain 3 (LC3I) conversion to the phosphatidylethanolamine-conjugated form of LC3I (LC3II) by Western blot. Results showed that levels of LC3 II gradually increased over time after treatment with EE-84 at 30 µM in K562 at 24, 48, and 72 h, respectively (Figure 6A). In addition, we observed the formation of cytoplasmic vacuoles in EE-84 treated K562 cells after 48 h at 30 μM by Diff-Quik staining, suggestive of autophagy induction. To validate the involvement of EE-84 in autophagy induction, we post-treated bafilomycin A1 (Baf-A1), an inhibitor of the vacuolar-type H^+^-ATPase which blocks the late phase of autophagy by preventing lysosomal acidification, at 40 nM for 8 h (cells were treated or not with EE-84 for 40 h before addition of Baf-A1 for 8 h), resulting in significant inhibition of vacuole formation after 48 h at 30 μM EE-84 (Figure 6B). To see the effect of autophagy inhibition on cell viability, we conducted a trypan blue assay test of EE-84 treated K562 cells with or without Baf-A1 post-treatment. 89.0% of cell viability was observed in EE-84 treated K562 cells with Baf-A1 post-treatment compared to 98.7% of cell viability in EE-84 treated K562 cells alone (Figure 6B).

In addition, studies have shown that autophagy and endoplasmic reticulum (ER) stress are closely related [8]. ER stress is considered a protective stress response in eukaryotic cells [15]. We then assessed whether EE-84 induces ER stress and subsequently regulates autophagy. We measured sensor proteins such as PERK, ATF6, and GRP78 that play a role in activating the unfolded protein response (UPR) in response to ER stress. PERK and phosphorylated eIF2*α* were significantly increased after treatment with EE-84 at 30 and 50 µM in K562 cells after 72 h. The abundance of ATF4, an effector of PERK and eIF2*α*, also increased, which indicated that the PERK-eIF2*α*-ATF4-CHOP pathway was activated under EE-84-induced ER stress (Figure 6C). Since GRP78 is involved in glycolysis [16], we then assessed the glycolytic flux levels in K562 after treatment with EE-84 at 50 μM for 72 h by using the Agilent seahorse XFp Glycolysis Stress test. The glycolysis and glycolytic capacity measured by the extracellular acidification rate (ECAR) significantly decreased after treatment with EE-84 compared to control (Figure 6D). We also confirmed the glycolytic flux levels in K562IR cells after treatment with EE-84 at 100 μM for 72 h. The results showed that the glycolytic capacity of ECAR also significantly decreased after treatment with EE-84 (Appendix A). Altogether, these data demonstrated that the prolonged treatment with EE-84 induced the PERK-eIF2*α*-ATF4-CHOP UPR pathway involved in EE-84-induced autophagy.

### 2.6. EE-84 Sensitized K562 Cells against Mcl-1 Inhibitor A-1210477 and Showed Synergistic Cytotoxicity in K562 and K562IR Cells

Conditions of ER stress may promote Mcl-1 protein stabilization via mechanisms involving UPR and ATF4 upregulation [17,18]. Considering the cytostatic potential but limited cytotoxicity of EE-84 along with the induction of ER stress, we then investigated the expression levels of the antiapoptotic protein, Mcl-1, which may be responsible for the apoptotic blockage. Based on the increase of antiapoptotic Mcl-1 expression in EE-84-treated K562 cells after 24 h compared to control (Figure 7A), we speculated that the combined treatment of the specific Mcl-1 inhibitor A-1210477 with EE-84 might sensitize K562 cells to apoptotic cell death. Using subtoxic concentrations of EE-84 (20 and 30 µM) and A-1210477 (10 µM), we first assessed the combinatory effects of these compounds using Hoechst/PI staining after 24 h (Figure 7B). We observed 46.89 ± 21.84% and 56.00 ± 25.46% induction of apoptotic cell death in combinatory treatments. The combination index (CI) of each compound-pair was calculated, and the combinatory treatment showed a synergistic effect (Figure 7C and Table 6). We also compared viability between EE-84 alone and in combination with A-1210477 after 24 h (Table 7). Our results showed that the Mcl-1-specific inhibitor A-1210477 sensitized EE-84-treated K562 and K562IR cells. Next, we confirmed the apoptotic cell death mechanism triggered by the combination treatment. As shown in Figure 7D, about 14.5% of apoptotic cell death was induced by A-1210477 alone, whereas over 40% of apoptosis was observed by a combination of EE-84 and A-1210477 after 24 h. z-VAD pretreatment completely prevented apoptosis induction in K562 cells in single and combined treatments (Figure 7D). The caspase-dependent nature of cell death modality was also confirmed by Annexin V APC/PI staining after 24 h in K562 cells (Figure 7E). Results showed that EE-84 (30 µM) in combination with A-1210477 (10 µM) induced 23.03 ± 4.08% early apoptosis (AnnV+/PI−) and 23.07 ± 5.32% late apoptosis (AnnV+/PI+) in K562 cells after 24 h of treatment (Table 8). Again, zVAD completely protected against cell death in all instances. Imatinib (1 μM) was used as a control. In addition, the cytotoxic effect of subtoxic concentrations of EE-84 (30 and 50 µM) in combination with A-1210477 (20 µM) was also tested on K562IR cells after 24 h of treatment, using the trypan blue exclusion test (Figure 7F). Results showed that viability was reduced by 25.4% when cells were cotreated with 20 µM A-1210477 and 30 µM of EE-84. A co-treatment with 50 µM EE-84 and 20 µM A-1210477 induced 40% of cell death. We further confirmed the combinatory effects of these compounds using Annexin V APC/PI staining after 24 h in K562IR cells (Figure 7G). Altogether, our results indicated that the specific Mcl-1 inhibitor A-1210477 sensitized K562 and K562IR cells to EE-84, significantly increasing cell death induction. Since mitochondrial dysfunction can trigger apoptosis, we measured the percentage of mitochondrial membrane potential (MMP) loss due to the combined effects of EE-84 with A-1210477 in K562 after 24 h. Results showed that the percentage of MMP loss significantly increased in K562 after cotreatment with EE-84 and A-1210477 (Figure 7H). In addition, we confirmed the combination effects of EE-84 with A-1210477 in K562 by colony formation assays (CFA); EE-84 combined with A-1210477 reduced the number of colonies, the total surface area, and the average size of colonies in K562 (Figure 7I). The obtained values for the two compound combinations denoted synergism, and 30 µM of EE-84 with 10 µM of A-1201477 were selected for further investigations.

### 2.7. Synergistic Induction of Caspase-Dependent Apoptosis by Combination Treatment of EE-84 and A-1210477 in K562 Cells

The cytotoxic effect and caspase-dependent apoptotic activity of the combination treatment were also validated by quantification of intracellular ATP levels. ATP levels decreased significantly after combination treatments compared to the compounds alone (Figure 8A). The 1 h pretreatment of z-VAD-fmk showed an increase in cell viability with the single treatment of 10 μM A-1210477 and the combined treatment. The addition of the caspase inhibitor showed no significant difference in the ATP levels for the single treatment of 30 μM EE-84. To further validate our data, the activity of caspase-3/7 was measured. There was a significant increase in the caspase-3/7 activity by 5.52-fold after the combined treatment of EE-84 and the Mcl-1 inhibitor compared to the untreated control (Figure 8B). To examine the detailed mechanism by which the combined treatment acts, we studied the intrinsic and extrinsic apoptotic pathways in K562 cells by Western blot (Figure 8C–E). First, the expression level of caspase-8, a cysteine protease that initiates apoptotic signaling via the extrinsic pathway, was detected. Cleavage of pro-caspase-8 in the combined treatment was observed (Figure 8C). Our results showed a reduction of pro-caspase-9, the initiator caspase critical for the intrinsic pathway, concomitant with the appearance of a cleaved fragment at 18 kDa after the combined treatment. In line with the previous observation of increased caspase-3/7 activity for the combination treatment, we observed a strong cleavage of pro-caspase-3 and cleaved fragments of caspase-7 for the combined treatment (Figure 8D). Our evaluation of antiapoptotic Mcl-1 and Bcl-xL expression levels showed an increase of both in EE-84 (30 μM) and A-1210477 (10 μM) single treatments in K562 cells as well as in the combined treatment (Figure 8E).

## 3. Discussion

Many products of marine organisms have been identified as modulators of cell death, exerting cytotoxic effects on cancer cells as activators of apoptosis, autophagy, or oncosis [19]. Currently, seven marine-based drugs have been approved, 23 compounds are undergoing phase I–III clinical trials, and thousands of compounds have been isolated from marine life and are undergoing preclinical studies [20]. Among these, four are used for cancer treatment: cytarabine (Cytosar-U^®^, Vitaris, Canonsburg, PA, USA), trabectedin (Yondelis^®^, PharmaMar, Colmenar Viejo, Madrid, Spain), eribulin mesylate (Halaven^®^, Eisai Inc., Tokyo, Japan) and the conjugated antibody brentuximab vedotin (Acentris^®^, Takeda Oncology, Cambridge, MA, USA) [21]. Cytosine arabinoside (cytarabine), originally isolated from the sponge *Cryptothethya crypa*, and now produced synthetically, is one of the most effective drugs to treat acute myeloid leukemia [21,22,23]. In addition, trabectedin, a marine metabolite of *Ecteinascidia turbinata*, is used for the treatment of soft tissue sarcoma [24]. However, despite the broad array of marine compounds clinically available or investigated, there is still an enormous library of natural products that remains untapped.

Aplysinopsins, a class of marine indole alkaloids, comprise two main distinct moieties—an indole and an imidazolidinone ring—and are isolated from a variety of marine organisms, including sponges [25], corals [26], anemone [27], and mollusks [28,29]. Aplysinopsins were first isolated by Kazlauskas et al., and initially identified as the major metabolite of eight Indo-Pacific sponge species of the genera *Thorecta* [12]. Since then, more aplysinopsin derivatives have been discovered and extracted from *Verongia spengeli*, *Dercitus* sp., *Smenospongia aurea*, *Verongular rigida*, *Dictyoceratida* sp., *Aplysinopsis reticulata*, *Aplysina* sp., *Hyrtios erecta*, and *Thorectandra* [28], among others. In line with the effort to identify new aplysinopsins with therapeutic potential, we report a set of synthetic aplysinopsin derivatives that possess anti-leukemic effects. In addition, aplysinopsin derivatives displayed a range of cytostatic effects, some inducing antiproliferating effects on myeloid leukemia cells more than others. We identified EE-84 as having the most potent antiproliferative effect in several chronic myeloid leukemia cell lines (K562, KBM5, MEG01, K562IR, and KBM5IR) as shown by trypan blue exclusion tests, CFAs, Hoechst/PI, and Annexin V/PI staining assays.

Identifying lead candidates from a library of compounds and characterizing their safety profiles by testing their toxicity in healthy models is essential in the drug discovery process. In addition, it helps predict clinical adverse effects in the future [28]. EE-84 showed differential toxicity in the noncancerous cell model, RPMI 1788, compared to K562, and was nontoxic to PBMCs. In the zebrafish model. EE-84 was also well-tolerated. The safe profile and drug-likeness of EE-84, supported by Lipinski’s rule of five, warranted further investigation of this compound.

EE-84 induced morphological changes in K562 cells as shown by diff-Quik staining and TEM imaging. In parallel with the cytostatic effect, EE-84 triggered a significant increase in the cell size together with the appearance of mitochondrial damage after 48 h of treatment. Furthermore, evaluation of mitochondrial dysfunction by Agilent Seahorse XFp Cell Mito Stress test showed significant inhibition of mitochondrial function after 72 h of 30 µM treatment. Interestingly, EE-84-treated K562 cells displayed a senescent-like phenotype, suggesting an interplay between mitochondrial dysfunction and cellular senescence in the EE-84-induced antiproliferative effect.

Cellular senescence is one of the many defense mechanisms that cells undergo to combat extrinsic and/or intrinsic stresses by halting cell cycle progression. Drug-mediated cellular stress leading to senescence is often accompanied by senescence-associated secretory phenotype (SASP). Many studies revealed that in senescent cells, mitochondrial function is significantly affected [30]. For example, Galanos et al. showed perturbations in the mitochondrial morphology in p21-inducible precancerous and cancerous cellular models (Li-Fraumeni and Saos-2 cell lines), characterized by the enlargement of the mitochondria and the damaged morphology of cristae [31,32]. Mitochondria were elongated or branched, and the cristae were abnormally distributed or lost, suggesting a disturbance of mitochondrial dynamics in the senescent cell. Wiley et al. also reported a distinct type of senescence associated with mitochondrial damage called “Mitochondrial Dysfunction Associated-Senescence” (MiDAS). They observed that mitochondrial dysfunction induces senescence and differs from the senescence caused by genotoxic or oncogenic stress by analyzing the secretome [33].

Senolytic compounds specifically induce apoptosis in senescent cells [34]. Dasatinib has been approved to treat CML as one of the second-generation tyrosine kinase inhibitors used in imatinib resistance and/or intolerance [35]. The senolytic drug combination of dasatinib and quercetin decreased senescent cells in human adipose tissue [36]. In addition, BH3 mimetic ABT-263 (Navitoclax) also induced apoptosis in mouse senescent bone marrow hematopoietic stem cells (HSCs) as a potent senolytic drug [37]. Hence, these studies support the idea that EE-84 triggered a senescent-like phenotype in response to mitochondrial dysfunction, autophagy, and prolonged ER stress as cellular metabolic alteration. In the present study, we showed that EE-84 induces a senescent-like phenotype. Interestingly, the combination of BH3 mimetic A-1210477 with EE-84 further triggers canonical apoptotic cell death in K562 and K562IR. We speculate that under these conditions, Mcl-1 inhibitor A-1210477 may act as a senolytic compound able to eradicate imatinib-sensitive and resistant CML cells. The effect of EE-84 as a single agent points at the ability of this compound to act as a general stress inducer. New effective combinatorial anticancer treatments also include stress inducers forcing cancer cells to rely more on a prosurvival factor. Targeting this specific protein promotes the eradication of cancer cells. Interestingly, some of the cases reported do involve prosurvival Bcl-2 family proteins. Dexamethasone, for example, is leading to Bcl-2 dependence and sensitization to venetoclax in multiple myeloma by altering the balance between pro- and antiapoptotic Bcl-2 protein members (increased Bim and decreased Mcl-1 vs. Bcl-2 upregulation), making Bcl-2 expression essential for survival [38]. Kapoor and colleagues report that CLL chronically resistant to ibrutinib (a Bruton tyrosine kinase (BTK) inhibitor) are sensitized to venetoclax partly also by a STAT3-mediated Bcl-2 upregulation in resistant CLL, implying a shift in the type of Bcl-2 family protein dependence [39]. At this level of investigation, we might speculate that EE-84 contributes to a similar scenario. Mcl-1 protein upregulation (or stabilization?) might signalize an increased reliance of CML cells on Mcl-1 as part of a stress response induced by EE-84. Further investigations will be required to elucidate any modulatory effects of EE-84 on the expression level of proapoptotic Bcl-2 proteins. In this view, EE-84 might be a compound to consider in combinatorial regimens.

Next, we investigated whether autophagy and ER stress occur concurrently with inhibition of proliferation by EE-84 in K562 cells. Accumulation of LC3-II, a standard marker for autophagosomes, was detected increasingly over time after 24, 48, and 72 h of EE-84 treatment, as seen in our Western blotting results. There was a significant increase in the formation of cytoplasmic vacuoles after 48 h EE-84 treatment at 30 μM, and this phenomenon was abrogated by a post-treatment with Baf-A1. Furthermore, 98.7% of K562 cells were viable after 48 h EE-84 treatment alone; however, the viability of K562 cells significantly decreased with Baf-A1 post-treatment, implying that EE-84 drives the K562 cells to undergo autophagy as a cell survival mechanism. ER stress was also activated by EE-84, given that sensor proteins of UPR, such as PERK and ATF6, and downstream effectors such as CHOP and ATF4 were upregulated to varying degrees. GRP78 has been initially characterized as a glucose-regulated protein [16]. Our Western blot results showed decreased levels of GRP78. These results are in line with data obtained by using the Seahorse glycolysis stress test. We observed reduced glycolysis levels, supporting the hypothesis of an impairment of the glycolytic function by EE-84 treatment in K562 and K562IR cells. Altogether, EE-84 induced autophagy, ER stress, and mitochondrial alterations as cell stress reactions, further leading to apoptosis. Overall, EE-84 plays a potential role in inhibiting CML cell activities via induction of cellular stress modalities.

The evasion of apoptotic cell death by cancer cells can impair responses to anticancer therapy. Prosurvival B-cell lymphoma 2 (BCL-2) proteins play a role of perpetrators in this scenario because they prevent apoptosis by keeping the cell death effectors like BAX and BAK under control [40]. BH3 mimetics offer a solution to this as they are designed to inhibit antiapoptotic BCL2 family proteins, leading to BAX and BAK activation, and thus promoting apoptosis [41]. Mcl-1 became a popular therapeutic target because it is one of the most frequently amplified genes across all human cancers. Moreover, an increase in Mcl-1 expression is commonly associated with chemotherapy resistance [42]. This study evaluated the synergistic effect of EE-84, a cytostatic marine compound, with the Mcl-1 inhibitor A-1210477 against CML K562 and K562 imatinib-resistant cells. We showed that the cotreatment of the marine compound and the Mcl-1 inhibitor induced apoptotic cell death along with the activation of caspase activity. These results are in line with other studies in which BH3 mimetics like ABT199 showed synergism with cell stress inducers like cardiac glycosides [43,44] and coumarin derivatives [44].

In the present study, we investigated the preclinical use of aplysinopsins as anti-leukemic agents. Our results identify EE-84 as a potential drug candidate for CML as it possesses drug-like properties and is well-tolerated in healthy models in vitro and in vivo. Mechanistically, EE-84 induces antiproliferative effects associated with the complex interplay between mitochondrial dysfunction, a senescent-like phenotype, autophagy, and ER stress, potentially inducing a condition favorable for senolysis by synergizing with senolytic compounds. To potentiate the anti-leukemic effects of EE-84, we suggest the cotreatment of this aplysinopsin derivative with the BH3 mimetic A-1210477 in K562 and K562IR, as it significantly increases cell death of malignant cells. In conclusion, the combination of EE-84 with BH3 mimetics is efficient and highly synergistic. Future investigations will determine whether this combinatory approach can become a therapeutic opportunity against resistant forms of CML.

## 4. Materials and Methods

### 4.1. Chemistry

#### 4.1.1. General Information

All reagents and solvents were of commercial grade. Melting points were determined on the digital melting point apparatus (Electro thermal 9100, Electro thermal Engineering Ltd., serial No. 8694, Rochford, UK) and are uncorrected. Elemental analyses were performed on a FlashSmart™ Elemental Analyzer (Thermo Scientific, Courtaboeuf, France) and were found within ±0.4% of the theoretical values.^1^H and ^13^C NMR spectra were measured with a Bruker Avance spectrometer (Bruker, Germany) at 400 and 101 MHz, respectively, using TMS as the internal standard. Hydrogen coupling patterns are described as (s) singlet, (d) doublet, (t) triplet, (q) quartet, and (m) multiplet. The chemical shifts were defined as parts per million (ppm) relative to the solvent peak. The reaction progress was checked by pre-coated TLC Silica gel 0.2 nm F254 nm [Fluka], visualized under UV lamp 254 and 365 nm. 2-Cyanoacetohydrazide [45]; N-benzyl indoles [46] methyl creatinine [47]; 5-methoxy indole-3-aldehyde [48] were prepared as reported. 1-(2-Amino-5-methyl-4,5,6,7-tetrahydrobenzo[b]thiophen-3-yl)ethan-1-one [49] was provided by Ahmed B. Abdelwahab, Ph.D., UMR CNRS 7565 SRSMC, Université de Lorraine, 57070 Metz, France.

#### 4.1.2. General Procedure for the Preparation of EE-31, EE-80, EE-84, EE-92

To a solution at 0 °C of oxalyl chloride (0.44 mL, 5.1 mmol) in dry ethyl ether (25 mL) was added dropwise a solution of indole (4.14 mmol) in dry ethyl ether (5 mL). The resulting solution was refluxed for 2 h. After removing the solvent under vacuo, the residue was dissolved in dry tetrahydrofuran (20 mL) and cooled to 0 °C. To the THF solution was added slowly the amine (9.73 mmol) in dry tetrahydrofuran (20 mL). After complete addition, 1 mL of triethylamine was added, and the reaction was left to stir overnight. The precipitate obtained was filtered off, washed several times with water, dried, and recrystallized from acetone.

2-(1-Benzyl-1H-Indol-3-yl)-N′-(2-Cyanoacetyl)-2-Oxoacetohydrazide EE-31

Yield 0.45g, 86%; mp 264–6 °C; ^1^H NMR (400 MHz, DMSO) δ: 10.63 (s, 1H), 8.90 (s, 1H), 8.26 (s, 1H), 7.72–7.50 (m, 5H), 7.41–7.21 (m, 4H), 5.46 (s, 2H), 3.83 (s, 2H); 13C NMR (101 MHz, DMSO) δ: 162.16, 141.47, 136.79, 136.70, 129.14, 128.97, 128.56, 128.26, 128.01, 127.73, 127.65, 127.29, 126.98, 124.17, 123.56, 122.53, 121.92, 116.04, 112.14, 111.95, 50.28, 24.10; Anal. Calcd for C_20_H_16_N_4_O_3_ (360.37): C, 66.66; H, 4.48; N, 15.55; found: C, 66.87; H, 4.50; N, 15.42.

2-(1-Benzyl-5-Methoxy-1H-Indol-3-yl)-N′-(2-Cyanoacetyl)-2-Oxoacetohydrazide EE-80

Yield (0.31g, 63%); mp 199–201 °C; ^1^H NMR (400 MHz, DMSO) δ: 10.62 (s, 1H), 8.78 (s, 1H), 7.76 (s, 1H), 7.57–7.43 (d, H), 7.41–7.20 (m, 5H), 7.06–6.77 (m, 2H), 5.44 (s, 2H), 3.87 (m, 3H), 3.33 (s, 2H); ^13^C NMR (101 MHz, DMSO) δ 161.34, 162.32, 156.47, 136.51, 131.22, 128.74, 127.83, 127.60, 127.33, 115.52, 113.07, 112.52, 103.72, 55.37, 50.06, 23.80; Anal. Calcd for C_21_H_18_N_4_O_4_ (390.40): C, 64.61; H, 4.65; N, 14.35; found: C, 64.55; H, 4.70; N, 14.32.

(N-(3-Acetyl-4,5,6,7-Tetrahydro-5-Methylbenzo[b]Thiophen-2-yl)-2-(1-Benzyl-5-Methoxy-1H-Indol-3-yl)-2-Oxoacetamide EE-84

Yield (0.38g, 61%); mp 165–7 °C; ^1^H NMR (400 MHz, CDCl_3_) δ: 13.58 (s, 1H), 9.07 (s, 1H), 8.08 (d, *J* = 2.5 Hz, 1H), 7.35–6.90 (m, 8H), 5.38 (s, 2H), 3.92 (d, *J* = 6.4 Hz, 3H), 2.82–2.80 (dd, *J* = 16.1, 5.0 Hz, 1H), 2.58–2.55 (m, 2H), 2.41 (s, 3H), 1.91–1.87 (m, 2H), 1.15 (d, *J* = 6.4 Hz, 2H), 1.10 (d, *J* = 6.6 Hz,3H); ^13^C NMR (101 MHz, CDCl_3_) δ: 196.38, 194.06, 164.12, 157.30, 140.94, 135.40, 131.26, 130.59, 130.42, 129.06, 128.98, 128.29, 128.24, 126.91, 123.08, 117.26, 115.78, 114.52, 111.48, 104.33, 55.81, 51.54, 36.56, 35.96, 31.01, 30.65, 29.25, 24.52, 21.76; Anal. Calcd for C_29_H_28_N_2_O_4_S (500.61): C, 69.58; H, 5.64; N, 5.60; S, 6.40; found: C, 69.60; H, 5.56; N, 5.60; S, 6.48.

N′-(2-Cyanoacetyl)-2-(1-(2,4-Dichlorobenzyl)-5-Methoxy-1H-Indol-3-yl)-2-Oxoacetohydrazide EE-92

Yield (0.25g, 56%); mp 242–4 °C; 1H NMR (400 MHz, DMSO) δ: 10.74 (s, 1H), 10.39 (s, 1H), 8.70 (s, 1H), 7.75 (dd, *J* = 17.6, 2.3 Hz, 2H), 7.58–7.26 (m, 2H), 7.09–6.83 (m, 2H), 5.65 (s, 2H), 3.82 (s, 3H), 3.34 (s, 2H); 13C NMR (101 MHz, DMSO) δ: 180.68, 161.35, 156.58, 140.97, 133.49, 133.29, 132.90, 131.21, 130.50, 129.22, 127.88, 113.29, 112.27, 111.62, 103.79, 55.39, 47.52, 23.80; Anal. Calcd for C21H16Cl2N4O4 (459.28): C, 54.92; H, 3.51; Cl, 15.44; N, 12.20; found: C, 54.93; H, 3.46; Cl, 15.54; N, 12.33.

#### 4.1.3. General Procedure for the Preparation of EE-115

1,3-Dimethyl creatinine (2.8 g, 22.7 mmol) and indole-3-aldehydes (22.7 mmol) were heated under reflux in piperidine (30 mL) for 4 h. After cooling, the reaction mixture was poured into water (200 mL) and then stirred for 30 min. The precipitate was filtered, washed several times with water, air dried and crystallized from methanol.

(Z)-5-((1H-Indol-3-yl)Methylene)-1,3-Dimethylimidazolidine-2,4-Dione (Aplysinopsin) EE-115 [50]

Yield 80%, mp 236–8 °C (reported mp 236 °C); ^1^H NMR (400 MHz, CDCl_3_) δ: 8.87 (d, *J* = 2.7 Hz, 1H), 8.54 (s, 1H), 7.85–7.66 (m, 1H), 7.45 (dd, *J* = 6.7, 1.4 Hz, 1H), 7.33–7.18 (m, 3H), 6.42 (s, 1H), 3.34 (s, 3H), 3.23 (s, 3H).

### 4.2. Cell Lines and Cell Cultures

The human chronic myeloid leukemia K562 (ATCC, CCL-243, Manassas, VA, USA), MEG01 (ATCC, CRL-2021, Manassas, VA, USA), and the normal B lymphocyte RPMI 1788 cell lines (KCLB, 10156, Seoul, Korea) were cultured in Roswell Park Memorial Institute (RPMI) 1640 medium (Lonza, Basel, Switzerland), supplemented with 10% heat-inactivated fetal bovine serum (FBS) (Biowest, Riverside, CA, USA) and 1% penicillin-streptomycin solution (100×) (GenDEPOT, Katy, TX, USA). KBM-5 cells were kindly donated by Dr. Bharat B. Aggarwal. Imatinib-resistant KBM5 cells (KBM5R) cells were obtained by sequentially increasing the concentration of imatinib from 0.25 to 1 µM imatinib in IMDM media supplemented with 10% (*v*/*v*) fetal calf serum and 1% (*v*/*v*) antibiotic–antimycotics [51]. Imatinib-resistant K562 (K562IR) cells were a gift of the Catholic University of Seoul and cultured in RPMI 1640 medium with 25 mM HEPES (Lonza) supplemented with 10% (*v*/*v*) FCS and 1% (*v*/*v*) antibiotic–antimycotics. Both resistant cell types were cultured with 1 µM of imatinib and washed three times before each experiment. Cells were maintained at 37 °C and 5% of CO_2_ in a humified atmosphere. Mycoplasma detection by Mycoalert^TM^ (Lonza) was performed every 30 days, and cells were used within three months after thawing.

Peripheral blood mononuclear cells (PBMCs) were isolated by density gradient centrifugation using Ficoll-Hypaque (GE Healthcare, Roosendaal, The Netherlands) from freshly collected buffy coats as previously described [51], obtained from healthy adult human volunteers (Red Cross, Luxembourg City, Luxembourg) after ethical approval as well as written informed consent from each volunteer. After isolation, cells were incubated overnight at 2 × 10^6^ cells/mL in RPMI 1640 (supplemented with 1% antibiotic–antimycotic and 10% FCS (BioWhittaker, Verviers, Belgium) at 37 °C and 5% CO_2_ in a humidified atmosphere. The day after, cell concentration was adjusted at 1 × 10^6^ cells/mL using the same fresh complete medium and then treated as indicated.

### 4.3. Compounds

Mcl-1 inhibitor, A-1210477 (S7790, Selleckchem, Seoul, Korea) was used in single and combination treatments. Etoposide (E1383, Sigma-Aldrich, Seoul, Korea), Imatinib (SML 1027, Sigma-Aldrich, St. Louis, MO, USA), and Celecoxib (PZ0008, Sigma-Aldrich, Seoul, Korea) were used as positive controls. Caspase inhibitor 1 (z-VAD-fmk, 187389-52-2, Calbiochem, Seoul, Korea) served to inhibit caspase-dependent apoptosis.

### 4.4. Cell Proliferation and Viability

Cell proliferation and viability were assessed by the trypan blue exclusion method (Lonza), and viable cells were counted using a hematocytometer (Marienfeld, Lauda-Königshofen, Germany). Differential toxicity was calculated by comparing the viability of RPMI 1788 cells to the viability of cancer cells (normal/cancer cells). The difference in viability was expressed in terms of fold change.

### 4.5. Colony Formation Assay

For 48 h pretreatment of EE-84 colony formation assays, approximately 3 × 10^5^ cells were seeded in each well of a 24-well plate, treated with EE-84 at indicated concentrations, and incubated at 37 °C and 5% of CO_2_ in a humidified atmosphere for 48 h. After 48 h, 1000 cells were counted and grown in a semisolid methylcellulose medium (Methocult H4230, StemCell Technologies Inc., Vancouver, BC, Canada) supplemented with 10% FBS. Colonies were detected after 10 days of culture by adding 1 mg/mL of 3-(4,5-dimethylthiazol-2-yl)-2,5-diphenyltetrazolium bromide (MTT) reagent (Sigma-Aldrich) and were analyzed by Image J 1.8.0 software (U.S. National Institute of Health, Bethesda, MD, USA).

### 4.6. Quantification of Apoptosis and Necrosis

The percentage of apoptotic cells was quantified as the fraction of cells showing fragmented nuclei, as assessed by fluorescence microscopy (Nikon, Tokyo, Japan) after staining with Hoechst 33342 (Sigma-Aldrich) and propidium iodide (Sigma-Aldrich). In addition, apoptosis was confirmed by Annexin V APC (Biolegend, 640919)/propidium iodide (BD Biosciences, 556547) staining and fluorescence-activated cell sorter (FACS) analysis according to the manufacturer protocol (BD Biosciences, 556547).

### 4.7. In Silico Drug Likeliness Properties

Lipinski’s ‘rule of five’ for drug likeliness properties was evaluated using the SCFBio website (http://www.scfbio-iitd.res.in/).

### 4.8. Zebrafish Toxicity

For toxicity assays, embryos were treated with 0.003% phenylthiourea (PTU) 14 h before the assay to remove pigmentation. Then, 2 h before the assay, the embryo’s shell was eliminated and then treated for up to 24 h with aplysinopsin compounds at indicated concentrations in 24-well plates. Viability and abnormal development were assessed after 24 h of treatment under light microscopy (Carl Zeiss Stereomicroscope DV4, Seoul, Korea). Pictures were taken by fixing embryos onto a glass slide with 3% methylcellulose (Sigma-Aldrich).

### 4.9. Cell Cycle Analysis

Cells were collected and fixed in 70% ethanol. DNA was stained with propidium iodide (PI) solution (1 μg/mL, Sigma-Aldrich, St. Louis, MO, USA) in 1XPBS (Biosesang, Seongnam, Korea) for 30 min at 37 °C, supplemented with RNase A (100 μg/mL, Roche, Basel, Switzerland). Samples were analyzed by flow cytometry using the FACSCalibur^TM^ system, Becton Dickinson (BD) Biosciences (San Jose, CA, USA). Data were recorded statistically (10,000 events/sample) using the Cell Quest software (BD Biosciences) and analyzed using Flow-Jo 8.8.5 software (Tree Star, Inc., Ashland, OR, USA).

### 4.10. Cell Morphology/Wright-Giemsa Staining

Diff Quik staining was used to analyze the morphological features of compound-treated cells. Approximately 3 × 10^5^ cells were seeded in each well of a 24-well plate and treated with EE-84 at the indicated concentrations for the indicated time. Cells were then spun onto a microscope glass slide for 5 min at 500× *g* using a cytopad with caps (Elitech Group Inc., Puteaux, France). Cells were fixed, air-dried, and then stained with the Diff-Quik staining kit (Sysmex, Kobe, Japan). The stained cells were examined, and images were captured with an inverted microscope (Nikon Eclipse Ti2).

### 4.11. SA-β-Gal Assay

The senescence-associated (SA)-β-Gal activity was measured as previously reported [52]. K562 cells treated with 80 nM doxorubicin 72 h were used as a positive control for senescence induction.

### 4.12. Analysis of Cell Size and Complexity (Granularity)

Flow cytometry acquisitions of FSC-H (forward) vs. SSC-H (size) were performed as a method to monitor the cell size and granularity (cell complexity) in untreated vs. EE-84-treated K562 (FACSCalibur^TM^; BD Biosciences). Data (10,000 events) were recorded using the Cellquest Pro software (BD Biosciences) and further analyzed using FlowJo software (Treestar, Ashland, OR, USA).

### 4.13. Analysis of Mitochondrial Membrane Potential (MMP) Levels

To monitor mitochondrial membrane potential (MMP), cells were incubated at 37 °C for 30 min with 50 nM MitoTracker Red CMXRos (all from Molecular Probes, Invitrogen, Grand Island, NY, USA) and then analyzed by flow cytometry. Data were recorded statistically (10,000 events/sample) using the CellQuest Pro software. Data were analyzed using the Flow-Jo 8.8.7 software, and results were expressed as cells with MMP loss (%).

### 4.14. Transmission Electron Microscopy (TEM)

Cells were pelleted and fixed in 2.5% glutaraldehyde (Electron Microscopy Sciences, Hatfield, PA, USA) diluted in 0.1 M sodium cacodylate buffer, pH 7.2 (Electron Microscopy Sciences) overnight. Cells were then rinsed twice with sodium cacodylate buffer, postfixed for 2 h in 2% osmium tetroxide at room temperature, washed with distilled water, and stained with 0.5% uranyl acetate at 4 °C overnight. Samples were then dehydrated in successive ethanol washes, followed by infiltration of 1 (100% ethanol):1 (Spurr’s resin). Samples were kept overnight embedded in 100% Spurr’s resin, mounted in molds, and left to polymerize in an oven at 56 °C for 48 h. Ultrathin sections (70–90 nm) were cut with an ultramicrotome, EM UC7 (Leica Microsystems Ltd., Seoul, Korea). Sections were stained with uranyl acetate and lead citrate and subsequently viewed using a JEM1010 transmission electron microscope (JEOL Korea Ltd., Seoul, Korea).

### 4.15. Determination of the Oxygen Consumption Rate and Glycolysis Stress

The oxygen consumption rate (OCR) was measured using a Seahorse XFp Cell Mito Stress Assay (#103010-100, Agilent Technologies, Seoul, Korea) ran on a Seahorse XFp analyzer (Agilent Technologies, Seoul, Korea) according to the manufacturer’s instructions. Briefly, cells were seeded at 30,000 cells per well and treated with EE-84 for 48 h in 175 μL medium. Before measurements, plates were equilibrated in a CO_2_-free incubator at 37 °C for 1 h. The analysis was performed using 1.5 µM oligomycin, 0.5 µM carbonyl cyanide-4-(trifluoromethoxy)phenylhydrazone (FCCP), and 1 µM rotenone/antimycin A as indicated. Data were analyzed using the Seahorse XF Cell Mito Stress Test report generator software (Agilent). The extracellular acidification rate (ECAR) was measured in response to the sequential injection of 10 mM glucose, 2 µM oligomycin (H+-ATP-synthase inhibitor), and 50 mM 2-deoxy-d-glucose (2DG) (hexokinase inhibitor) to detect non-glycolytic acidification, glycolysis, maximal glycolytic capacity, and glycolytic reserve using a Seahorse XFp analyzer with a Glycolysis Stress Test kit (Agilent, Santa Clara, CA, USA).

### 4.16. Measurement of Intracellular ATP Content

To quantify metabolically active cells, intracellular ATP levels were measured by CellTiter-Glo Luminescent Cell Viability Assay (Promega, Cosmogenetech, Seoul, Korea) following the manufacturer’s protocol.

### 4.17. Measurement of Caspase-3/7 Activity

According to the manufacturer’s instructions, the activation of caspase-3/7 was measured using the Caspase-Glo 3/7 Assay (Promega, Madison, WI, USA). The caspase-3/7 reagent was added to the sample volume at a 1:1 ratio, and the cells were incubated for 1 h at room temperature. The luminescence of triplicate samples was measured using a microplate reader.

### 4.18. Whole-Cell Extracts and Western Blotting

For the preparation of whole-cell extracts, cells were harvested, washed in cold 1xPBS, and lysed in Mammalian Protein Extraction Reagent (M-PER^TM^, Thermo Fisher, Waltham, MA, USA) supplemented with a 1× protease inhibitor cocktail (Complete, EDTA-free, Roche, Basel, Switzerland) according to the manufacturer’s instructions. Protein concentration was measured using the Bradford assay. Proteins were aliquoted and stored at −80 °C. Afterward, proteins were subjected to sodium dodecyl sulfate (SDS)–polyacrylamide gel electrophoresis (PAGE) and transferred to PVDF membranes (GE Healthcare, Little Chalfont, UK). Membranes were incubated with selected primary antibodies: anti-caspase-7 (9494S), anti-caspase-9 (9502S), anti-caspase-8 (9746), anti-Mcl-1 (4572S), anti-LC3B (2775), ATF4 (11815), ATF6 (65880), Bip/GRP78 (C50B12), CHOP (L63F7), eIF2α (9722) and PERK (3192) from Cell Signaling (Danvers, MA, USA), anti-caspase-3 (sc-56053) and anti-PARP-1 (sc-53643) from Santa Cruz Biotechnology (Dallas, TX, USA), anti-Bcl-xL (610212) from BD Pharmingen (San Jose, CA, USA), eIF2a (Phospho-Ser51) (11279) from Signalway Antibody Co. (College Park, MD, USA) and anti-β-actin (5441) from Sigma Aldrich (St. Louis, MO, USA). Blots were probed in PBS-T containing the appropriate blocking agent (5% milk or 5% BSA) for 1 h. Membranes were prehybridized overnight with the indicated primary antibodies. After washing, blots were incubated with species-appropriate HRP-conjugated secondary antibody (Santa Cruz) in PBS-T containing 5% milk. Proteins of interest were detected with ECL Plus Western blotting Detection System reagent (GE Healthcare) using ImageQuant LAS 4000 mini system (GE Healthcare).

### 4.19. Statistical Analysis

Data are expressed as the mean ± SD and significance was estimated by using one-way or two-way ANOVA (analysis of variance) followed by either Dunnett’s multiple comparison test or Sidak’s multiple comparison test, unless otherwise stated, using Prism 8 software, GraphPad Software (La Jolla, CA, USA). *p*-values were considered statistically significant when *p* < 0.05. Legends are represented as follows: * *p* < 0.05, ** *p* < 0.01, *** *p* < 0.001.

## 5. Conclusions

Through the screening of several aplysinopsin analogs, we selected EE-84 as an interesting anti-leukemic agent. EE-84 exhibited a safety profile as it had minimal impact on healthy models in vitro and in vivo. We also found that the treatment of K562 with EE-84 induced an antiproliferative effect concomitant with autophagy and ER stress induction as well as senescence. In addition, mitochondrial dysfunction was observed in line with altered K562 cell morphology after EE-84 treatment. Treatment of EE-84 combined with the BH3 mimetic A-1210477 (specific for Mcl-1) potentialized apoptotic cell death. We suggest this cotreatment as a promising preclinical approach to therapeutic failure, specifically in resistant CML.

## Data Availability

Data supporting reported results can be found here: https://data.mendeley.com/datasets/z2rkrgwbm9/2; https://data.mendeley.com/datasets/bs3r484nfn/1; https://data.mendeley.com/datasets/ym9mcsm2fg/1, accessed on 19 May 2021.

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
