# Peer review of "Anti-Leukemic Properties of Aplysinopsin Derivative EE-84 Alone and Combined to BH3 Mimetic A-1210477"

_marinedrugs, 2021, doi:10.3390/md19060285_

Round 1

Reviewer 1 Report

The authors tested the anti-leukemic activity of Aplysinopsins analogues in leukemia models, focusing in particular on the EE-84 analogue due to the higher antiproliferative efficacy observed.

Compared to the previous version, other cell lines from CML were included in the study, such as some clones resistant to Imatinib, and other experiments were performed including the evaluation of the combination with the BH3 mimetic, A-1210477. However, the different cellular models of CML, resistant and non-resistant, were used only for CFA. Only the K562IRs were used in comparison to sensitive clone to evaluate the effects of the combination with A-1210477.

Since in this second submission the authors extended the evaluation of the effects of Aplysinopsin analogs only to CML models and not to AML, I would suggest deleting the U937 data, no more useful in this latest version of the manuscript.

There are many figures, but few numerical values in the text and in the figures that do not facilitate to quantize the phenomena described, especially regarding the levels of apoptosis and cell cycle changes. In particular, for each AnnV dot plot it would be useful to report the percentage of AnnV+/PI- and AnnV+/PI+ cells. Similarly, regarding the cytostatic effect of EE-84 in K562 (Figure 3c), it would be useful to indicate for each column the mean values +/- SD.

Overall, no advantage of EE-84 over imatinib emerges in imatinib-sensitive CML models, thus the question is whether the use of EE-84 alone or in combination with a BH3 mimetic may have a rationale in resistant models. This is not clear from the results shown.

Some of the major considerations are:

1) The authors show an increase in Mcl-1 following exposure to EE-84 and that the combination with A-1210477 could be effective in countering the apoptosis blockage due to EE-84-mediated Mcl-1 up-regulation. The combination, however, doesn't seem very effective even in the K562IR model (AnnV + around 30%?). What happens if EE-84 is used in combination with Imatinib instead? Does it increase the sensitivity to apoptosis of imatinib-resistant CML clones? Moreover, I'm concerned about the EE-84-mediated increased expression of Mcl-1, since Mcl-1 overexpression has usually been reported as a key factor in mediating drug resistance in leukemia.

2) Line 460-463 In the conclusions the authors state that… .. the combination of BH3 mimetic A-1210477 with EE-84 further triggers canonical apoptotic cell death in K562 and K562IR. Interestingly, under these conditions, Mcl-1 inhibitor A-1210477 acts as a senolytic compound able to eradicate imatinib-sensitive and resistant CML cells. … ..However these conclusions do not seem to me supported by the results shown in the work but rather seem overestimated.

In addition:

Line 89 - "only weekly affected proliferation" I suppose it should be replaced with "only weakly affected proliferation"

Materials and methods section:

- in the Cell Lines and Cell Cultures paragraph there is no reference to the other cell lines included in the new version of the manuscript.

- in the Quantification of Apoptosis and Necrosis paragraph there is no reference to Annexin V staining

- in the Statistical Analysis paragraph no mention of Dunnett's multiple comparisons test or Sidak's multiple comparisons test

- GI50 definition is missing.

Reviewer 2 Report

In manuscript entitled “Anti-leukemic properties of aplysinopsin derivative EE-84 alone and combined to BH3 mimetic 1210477” describe differential effect of EE-84 on leukemic cell lines vs. healthy cells and the combined effect of EE-84 with BH3 mimetic A-1210477. Out of several tested derivatives of aplysinopsin they found EE-84 to display the cytostatic effect on leukemic cell lines. They have shown that the chemical is not effective on healthy cells. In leukemic cells the compound induces several effects affecting UPR in ER, autophagy, mitochondrial metabolism. They found that BH3 mimetic A-1210477, effectively kills cells treated EE-84. Both compounds acting on cells synergistically. The combination of these drugs is thus a good candidate for preclinical test.

The experiments in the study are reasonably designed and well executed. I did not found any significant flaws. The results are properly interpreted and discussed. As far as I can judge, the manuscript is written in good language.

I therefore recommend to accept the manuscript for the publication in Marine drugs as it is, with the exception of  few minor mistakes, including incomplete references 12 and 19).

I also think, that title of the paper should be corrected by referring to the BH3 mimetic as to A-1210477, instead of just 1210477.

Reviewer 3 Report

Study described in the submitted manuscript aimed to determine the antileukemic activity of novel aplysinopsin derivative, EE-84. Additionally, the study evaluated the synergistic cytotoxic effect of EE-84 with the Mcl-1 inhibitor, A-1210477 on chronic leukemia K562 cell line. The submitted manuscript provides a large amount of data that are presented in a comprehensive manner. Authors provided insights on the possible mechanisms of EE-84 action in chronic leukemia cells, when used alone and in combination with A-1210477. This study, in general, puts together a set of important data that will be of interest to the wide anti-cancer therapy community. 

Comments/suggestions:

  1. In the title: „BH3 mimetic 1210477” the letter „A” in the name of inhibitor is missing.
  2. Sections 4.4, 2.1 - using trypan blue staining we can analyze the cell viability, but using hemocytometer we can estimate the number of cells, not their prolifaration (we are able to show the overall growth of cells but not directly their proliferation). The reduction in cell number, found by counting them in hemocytometer, can be caused by the inhibition of cell proliferation, cell cycle disturbances, and also cell death. Therefore, using the term „proliferation” is inaccurate giving the method applied. Why the NMR spectrum data are provided in 2.1 secton?

  3. I think at the beginning the authors should provide the information on the physical and chemical properties of the tested agents (eg NMR, drug-likeness) and then the biological effects and mechanisms of action. In the present form in some places the reader must jump from one issue to another and that is quite confusing (it applies to sections: Material and methods and Results).

  4. The Table S1 is not mentioned in the text. Moreover, In Table S1, the IC50 of aplysinopsin analogs in the non-cancerous cell line RPMI 1788 is provided whereas in the legend (line 701): „ Table S1: IC50 of aplysinopsin analogs in myeloid leukemia cell lines” is written. Additionally, why did Authors used the GC50 value for leukemia cells and IC50 value for normal cells?. Please clarify this difference. Additionally, please explain why GC50 estimation was used after trypan blue staining as GC50 refers to cytostatic not cytotoxic agents and trypan blue is used for analysis of cell viability not proliferation.

  5. Section 4.1: KBM5, MEG01 and resistant K562IR and KBM5IR CML cells are mentioned. There is no information about these cell lines in Material and method section. This also applies to PMBC.

  6. Why the section 4.12 was written seperatly if no specific methods with the use of cytometry were described in this section. Eg. The cell cycle and mitochondrial membrane potential were desribed in other sections. Moreover, the AnnexinV / PI assay is not mentioned in the material and method section.

  7. Section 4.13: why LysoTracker was used for MMP analysis? What are the results of LysoTracker analysis? Moreover, Authors wrote that (MMP) „results were expressed as mean fluorescence intensity (MFI)” whereas on Figure 7F MMP% is provided. Please clarify the method of MMP analysis.

  8. The results showed on Figure 7D are described two times in the text (section 2.6, line 298 and the following section also numbered 2.6- line 358).

  9. There is no information about the source of antibodies for Western blotting.

  10. Section 2.5: why did Authors choose to use the bafilomycin A1 for 8 h after 40h of cells treatment with EE-84? And not the co-treatment or after a shorter period of incubation with the tested agent?

  11. Section 2.5 line 244: „To distinguish the mode of autophagy, we conducted a trypan blue assay test of EE-84 treated K562 cells with or without Baf-A1 post-treatment”  - the use of trypan blue assay to distinguish the mode of autophagy is a little confusing.

  12. Section 2.6 lines 290-293: the increase of apoptosis after combination treatment should be compared especially to the values obtained in single-treatment groups, not only to control.

  13. Figure 7B – using Hoechst/PI staining we can’t discriminate early and late  apoptosis. Hoechst staining allows for observation of nucleus fragmentation and this is the late event in apoptosis. On Figure 3 Authors used other terms for cell population in Hoechst/PI analysis. It should be unified throughout the text.

Round 2

Reviewer 1 Report

The authors have addressed main points of criticism. However, some observations require to be considered in the text.

In the cover letter, the authors argue that "the primary focus of our study was to find alternative strategies for the management of CML to combat chemoresistance against Imatinib as mentioned in our introduction". Considering this, to determine if EE-84 in combination with A-1210477 is a viable alternative strategy, the most useful cell models are the resistant ones. The data in this regard, however, show that EE-84 alone is not effective and in combination it is no more effective than Imatinib particularly in resistant models (Fig S6 b). Thus, to avoid missing the "primary focus", I suggest the authors review the text trying to be cautius in indicating the combination as an alternative strategy in Imatinib resistance models. In particular, please reformulate line 522-524. Furthermore, I would move the data obtained on the resistant clone, K562IR, to the main text and the data of the K562 to the supplementary material.

In the cover letter the authors report that they “selected the imatinib-resistant cell line for combination treatments to assess the capacity of EE-84 to re-sensitize these cells to imatinib", but there are no data in the text showing the capacity of EE-84 to make cells sensitive again to imatinib. Please clarify this issue.

Fig 7: Panel F is missing, while panel E is repeated twice.

Line 65 and 439: add “Chronic” to myeloid leukemia cell lines.

Line 307 and line 597: Please change K562R whit K562IR

Reference 51 does not refer to the resistant line harboring Bcr-Abl-T315I mutation. Please clarify, review the information on the K562IR and replace the reference with the appropriate one.

Line 476-485: The two rebuttal references (#38 and #39) report an enhances mitochondrial Bcl-2 dependence which is a different concept than the simple increase of Bcl-2. In particular, Matulis et al state that “Co-treatment of human myeloma cell lines and primary patient samples, with Dex and venetoclax with Dex and venetoclax, significantly increased cell death over venetoclax [….]. The mechanism by which this occurs is an increase in the expression of both Bcl-2 and Bim upon addition of Dex. This results in alterations in Bim binding to anti-apoptotic proteins. Dex shifts Bim binding towards Bcl-2 resulting in increased sensitivity to venetoclax”. Similarly, Kapoor et al, citing Deng's original work, report that “CLL cells are sensitized to venetoclax by increasing mitochondrial dependence on BCL-2. BTK inhibition can achieve this in CLL cells by increasing BIM levels and decreasing the abundance or function of MCL-1”. The authors must speculate on this concept and reformulate this part of the discussion.
